# Mitogen Kinase Kinase (MKK7) Controls Cytokine Production In Vitro and In Vivo in Mice

**DOI:** 10.3390/ijms22179364

**Published:** 2021-08-29

**Authors:** Amada D. Caliz, Hyung-Jin Yoo, Anastassiia Vertii, Ana C. Dolan, Cathy Tournier, Roger J. Davis, John F. Keaney, Shashi Kant

**Affiliations:** 1Division of Cardiovascular Medicine, Department of Medicine, Brigham and Women’s Hospital, Harvard Medical School, Boston, MA 02115, USA; acaliz@partners.org (A.D.C.); hyoo@bwh.harvard.edu (H.-J.Y.); acdolan@bwh.harvard.edu (A.C.D.); jfkeaney@bwh.harvard.edu (J.F.K.J.); 2Department of Molecular, Cell and Cancer Biology, University of Massachusetts Medical School, Worcester, MA 01655, USA; anastassiia.vertii@umassmed.edu; 3Division of Cancer Sciences, School of Medical Sciences, Faculty of Biology, Medicine and Health, University of Manchester, Manchester M13 9PL, UK; cathy.tournier@manchester.ac.uk; 4Program in Molecular Medicine, University of Massachusetts Medical School, Worcester, MA 01655, USA; roger.davis@umassmed.edu

**Keywords:** kinase, MAPK, cytokine, signaling, inflammation

## Abstract

Mitogen kinase kinase 4 (MKK4) and mitogen kinase kinase 7 (MKK7) are members of the MAP2K family that can activate downstream mitogen-activated protein kinases (MAPKs). MKK4 has been implicated in the activation of both c-Jun N-terminal kinase (JNK) and p38 MAPK, while MKK7 has been reported to activate only JNK in response to different stimuli. The stimuli, as well as the cell type determine which MAP2K member will mediate a given response. In various cell types, MKK7 contributes to the activation of downstream MAPKs, JNK, which is known to regulate essential cellular processes, such as cell death, differentiation, stress response, and cytokine secretion. Previous studies have also implicated the role of MKK7 in stress signaling pathways and cytokine production. However, little is known about the degree to which MKK4 and MKK7 contribute to innate immune responses in macrophages or during inflammation in vivo. To address this question and to elucidate the role of MKK4 and MKK7 in macrophage and in vivo, we developed MKK4- and MKK7-deficient mouse models with tamoxifen-inducible Rosa26 Cre^ERT^. This study reports that MKK7 is required for JNK activation both in vitro and in vivo. Additionally, we demonstrated that MKK7 in macrophages is necessary for lipopolysaccharide (LPS)-induced cytokine production, M1 polarization, and migration, which appear to be a major contributor to the inflammatory response in vivo. Conversely, MKK4 plays a significant, but minor role in cytokine production in vivo.

## 1. Introduction

The mitogen-activated protein kinases (MAPKs) are an evolutionarily conserved signaling mechanism that controls a range of important cellular functions, including differentiation, stress response, and apoptosis [1,2]. Stress-activated MAPK is comprised of two major pathways, c-Jun N-terminal kinase (JNK) and p38 MAP kinase (p38) [3]. JNK and p38 are well-studied signaling cascades that are essential for vital cellular activities [4,5]. The activation of MAPK JNK and p38 are mediated by several members of the evolutionary conserved upstream mitogen-activated protein kinases kinase (MAP2K) family [6,7,8]. Such diversity in the repertoire of upstream activators might, in turn, define the specificity of the stimuli-induced pathway activation.

Both mitogen kinase kinase 4 (MKK4) and mitogen kinase kinase 7 (MKK7) are MAP2K family members. While MKK4 has been implicated in the activation of both downstream stress-activated MAPKs, JNK and p38 [3,4,9,10], MKK7 mainly activates the downstream kinase JNK in the presence of different stimuli [3,4]. The cell type and activation stimuli determine whether MAP2K, MKK7 or MKK4, plays a significant role in the downstream targets of JNK and p38 activation in vitro and in vivo [11,12,13].

The innate immune response is an evolutionarily conserved response and the first line of defense against harmful pathogens [14]. The inflammatory response is a well-orchestrated event initiated by pathogens, such as bacteria, and mediated by a quick activation of MAPK, leading to the production of cytokines. This MAPK activated cytokine production leads to the amplification of an inflammatory response that affects both the immune and non-immune cells [15]. Understanding the cell-type- and stimuli-specific mechanisms responsible for the signal transduction is essential for managing the uncontrolled inflammatory responses. The precise contribution of MKK4 and MKK7 in the macrophage-specific immune response has not been discerned. Using both in vivo and in vitro models, we determined the contribution of each kinase with respect to downstream target activation, cytokine secretion, and macrophage migration, all essential features of an effective immune response.

Pro-inflammatory cytokines, such as tumor necrosis factor-alpha (TNFα) and bacterial lipopolysaccharides (LPS), activate MAPKs, JNK, and p38 [16]. However, it is unclear which MAP2K plays a more prominent role during TNFα and LPS activation of JNK and p38 in macrophages. The purpose of this study is to determine whether MKK4 or MKK7 plays a critical role in JNK activation and cytokine production during inflammation in macrophages. In this study, we have shown that MKK7 plays a major role in TNFα and LPS-activated JNK signaling and cytokine production in vitro and in vivo. Although MKK4 contributes to LPS mediated inflammatory response in macrophages, its effects on the JNK signaling and cytokine production are modest compared to MKK7. Data from this study indicate that the MKK7 pathway represents a potential target for developing therapeutic drugs that may be beneficial for the treatment of sepsis and inflammation.

## 2. Results

### 2.1. Conditional MKK7 Mice Generation

JNK kinases can be activated by different MAP2Ks, which include MKK4 and MKK7 proteins [2,17]. p38 MAPK can be activated mainly by MKK3 and MKK6, but MKK4 can play a role in some circumstances [9]. However, the role of MKK7 in the activation of p38 MAPK is not known. Previously, we have established the role of mixed lineage kinases (MLKs), a member of mitogen-activated protein kinases kinase kinase (MAP3K) family in TNFα and LPS-activated JNK and p38 MAPKs in vitro and in vivo [4,16,18]. MLKs also regulate cytokine production in vitro and in vivo [4,16]. Additionally, activation of JNK and p38 via MKK4 and MKK7 have been implicated in cytokine production [11]. Nonetheless, the roles of MKK4 and MKK7 in LPS-induced macrophages and inflammation are not known. To investigate the role of MKK4 and MKK7 in LPS-induced cytokine production in vivo, we used our previously published MKK4 conditional mice [19] and generated MKK7 conditional mice (Appendix A). More specifically, the temporal deletion in MKK4 and MKK7 mice was achieved by crossing MKK4-floxed and MKK7-floxed mice with inducible Rosa26-Cre^ERT^ mice (Appendix A). Genotypic analysis utilizing different tissues isolated from tamoxifen-treated Rosa26-Cre^ERT^-*Mkk4*-floxed (*Mkk4*^∆/∆^) and Rosa26-Cre^ERT^-*Mkk7*-floxed (*Mkk7*^∆/∆^) mice demonstrated the specific disruption of the *Mkk4* and *Mkk7* genes (Appendix A), which was not observed in control mice. This finding was confirmed by immunoblot analysis of MKK4 and MKK7 expression in mouse embryonic fibroblasts (MEFs) isolated from control or *Mkk4*^∆/∆^ and *Mkk7*^∆/∆^ mice treated with 4-hydroxytamoxifen (Appendix A).

### 2.2. MKK7 Is Required for TNFα-Stimulated MAP Kinase Activation

A pro-inflammatory cytokine TNFα binds to TNFα receptors (TNFR) on the cell surface which activates the signal transduction mechanisms leading to the phosphorylation of MKK4 and MKK7 protein kinases [11]. We used primary mouse embryonic fibroblast cells (MEFs) treated with TNFα to examine the contribution of MKK4 and MKK7 to JNK and p38 MAPK activation. First, we isolated MEFs from control or Rosa26-Cre^ERT^-*Mkk4*-floxed and Rosa26-Cre^ERT^-*Mkk7*-floxed mice. These isolated MEFs were treated with 4-hydroxytamoxifen for specific gene deletion. Using these MKK4 and MKK7 knockout MEFs, we assessed the contribution of MKK4 and MKK7 in MEFs by stimulating these cells with TNFα and compared them to Rosa26 Cre^ERT^ control cells. We found that a deficiency of MKK4 caused only minor changes in TNFα-stimulated MAP kinase activation of JNK (Figure 1A). Conversely, TNFα-stimulated MAP kinase activation of JNK in *Mkk7*^∆/∆^ MEFs was markedly reduced (Figure 1A).

Next, bone marrow-derived primary macrophages (BMDMs) isolated from Rosa26-Cre^ERT^-*Mkk4*-floxed and Rosa26-Cre^ERT^-*Mkk7*-floxed conditional mice were treated with 4-hydroxytamoxifen for respective gene deletion. Similar to the MEFs, treatment with TNFα to MKK4-deficient BMDMs caused only minor changes in TNFα-stimulated JNK activation (Figure 1B). However, TNFα-stimulated JNK activation was severely affected by MKK7 deficiency compared to the controls (Figure 1B). We also observed a minor change in the activation of p38 MAPK in both MKK4 and MKK7-deficient BMDMs. Together, these data indicate that the majority of TNFα-stimulated MAP kinase activation is mediated by MKK7, while MKK4 acts as a minor partner.

### 2.3. MKK7 Plays an Important Role in LPS-Induced MAP Kinase Activation and Cytokine Production

To further examine the individual contributions of MKK4 and MKK7 in LPS-induced MAPK activation in macrophages, we compared the responses of JNK and p38 activation in wild-type control, *Mkk4*^∆/∆^ and *Mkk7*^∆/∆^ BMDMs to LPS treatment. Our data showed that, similar to TNFα-induced MEFs, MKK4 deficiency produces only a moderate decrease in JNK and p38 activation in LPS-induced macrophages (Figure 2). In contrast, LPS-induced MKK7-deficient macrophages showed a significant reduction in both JNK and p38 activation (Figure 2).

Next, we compared the production of inflammatory cytokines during LPS treatment in control, MKK4, and MKK7-deficient BMDMs. We found that both LPS-treated MKK4 and MKK7- deficient BMDMs secrete markedly less TNFα and interleukin 6 (IL6) than the wild-type control cells (Figure 3A). However, the reduction in cytokine production was significantly higher in *Mkk7*^∆/∆^ than *Mkk4*^∆/∆^ (Figure 3A). We also analyzed mRNA production of pro-inflammatory cytokines TNFα and IL6 in both MKK4 and MKK7- deficient BMDMs. Similar to the secreted cytokines proteins, mRNA expression was compromised in both MKK4 and MKK7-deficient BMDMs when compared to control cells. Consistent with cytokine secretion, the effect of MKK7-deficiency caused a greater decrease in TNFα and IL6 cytokine expression than MKK4-deficiency in BMDMs (Figure 3B). Collectively, these data show that MKK4 and MKK7 are required for cytokine production in LPS-induced macrophages, with MKK7 having a stronger impact on pro-inflammatory cytokines at the mRNA and protein levels.

### 2.4. MKK 7 Controls Cell Migration

MAPK signaling has been implicated in macrophage migration and invasion. Healthy macrophages require robust migration and invasion to execute their proper function; thus, highlighting the importance of determining the role of MKK4 and MKK7 in macrophage migration and invasion. We found that MKK4 deficiency leads to a modest disruption of cell migration and invasion in macrophages (Figure 4), while MKK7-deficiency produced a major blockage in the migration and invasion of macrophages compared to the control (Figure 4). These data demonstrate that MKK7 is a key component of macrophage migratory machinery.

### 2.5. MKK7 Contributes to Inflammatory Cytokine Production In Vivo

To examine the contribution of MKK4 and MKK7 to the immune response in vivo, we tested the effects of MKK4 and MKK7 deficiency on the response of mice to endotoxin exposure. Control Rosa26-Cre^ERT^, Rosa26-Cre^ERT^-*Mkk4*-floxed and Rosa26-Cre^ERT^-*Mkk7*-floxed conditional mice were injected with a single dose of tamoxifen for respective gene deletion. After 2 weeks of gene deletion, all three group of mice injected with endotoxin LPS. Treatment of wild-type mice with LPS caused increased production of inflammatory cytokines TNFα, IL6, IL1α, and IL1β in the blood (Figure 5). Although, the production of LPS-induce pro-inflammatory cytokines TNFα, IL6, IL1α, and IL1β were significantly suppressed in both *Mkk4*^∆/∆^ and *Mkk7*^∆/∆^ mice in comparison to wild-type control mice (Figure 5). 

However, this decreases in cytokine production in both *Mkk4*^∆/∆^ and *Mkk7*^∆/∆^, mice were significantly lower compared to the control, the decrease in cytokines TNFα, IL6, IL1α, and IL1β production in MKK4-deficient BMDMs were less compared to MKK7-deficient mice (Figure 5). Together, these data establish the MKK4 and MKK7 pathways as contributors to inflammation and, like isolated macrophages in vitro, suggested that MKK7 has a significantly stronger contribution to the LPS response than MKK4 in vivo.

### 2.6. MKK7 Is Required for M1 Macrophage Polarization

Macrophage polarization plays an important role during inflammation [20]. The macrophage can polarize into inflammatory (M1) or anti-inflammatory (M2) macrophages according to their microenvironment and perform different functions [20,21]. To determine if MKK7 is required for macrophage polarization, we performed an M1 and M2 polarization experiment in controls, *Mkk4*^∆/∆^ and *Mkk7*^∆/∆^ BMDMs. After polarization to either M1 and M2 macrophages, we performed RT-qPCR to check the expression of markers for M1 (TNFα) and M2 (IL6) macrophages in isolated BMDM cells of the controls, *Mkk4*^∆/∆^ and *Mkk7*^∆/∆^ mice. Results from RT-qPCR analysis demonstrated that macrophage-specific MKK7-deficiency decreased the expression of M1 marker genes (TNFα) in comparison to the controls (Figure 6A). Moreover, gene expression associated with M2 polarization (IL10) demonstrated that macrophage-specific MKK7-deficiency increased the expression of genes associated with M2 polarization (Figure 6B). These data confirm that MKK7 expression in macrophages promotes M1 polarization and suppresses M2 polarization.

### 2.7. JNK1/2 Mimics the Phenotype of MKK7 during Inflammatory Cytokine Production In Vitro

JNK signaling is severely affected by MKK7- deficiency (Figure 1). To dissect the role of JNK1/2 in macrophages, we isolated BMDMs from control Rosa26-Cre^ERT^ and Rosa26-Cre^ERT^-*Jnk1*-floxed *Jnk2−/−* mice and treated them with 4-hydroxytamoxifen for gene deletion (*Jnk1*^∆/∆^*Jnk2^−/−^*). These control Rosa26-Cre^ERT^ and *Jnk1*^∆/∆^*Jnk2^−/−^* macrophages were treated with LPS for 24 h. Similar to MKK7, LPS-treated JNK1/2- deficient BMDMs secrete markedly less TNFα and IL6 than wild-type cells (Figure 7A).

Thereafter, we tested the mRNA expression of the pro-inflammatory cytokines, TNFα and IL6, in JNK1/2- deficient BMDMs. We found that similar to the secreted cytokines, mRNA expression of TNFα and IL6 were reduced in JNK1/2- deficient BMDMs compared to the control cells (Figure 7B). These data suggest that JNK1/2 is required for cytokine production in LPS-induced macrophages.

Next, the role of the MKK7 downstream target, JNK, during the migration and invasion of macrophages was investigated. We used macrophages isolated from control and *Jnk1*^∆/∆^*Jnk2^−/−^* for these experiments. Similar to MKK7-deficiency, JNK1/2-deficiency in macrophages leads to a significant reduction of macrophage migration and invasion (Figure 7C). These data show that JNK1/2-deficient macrophages mimic the phenotype of MKK7 during macrophage migration.

Going forward, we also investigated the role of the JNK during M1 and M2 polarization of macrophages by using the macrophages isolated from control and *Jnk1*^∆/∆^*Jnk2^−/−^* mice. Similar to MKK7-deficiency, JNK1/2-deficiency in macrophages leads to a significant reduction of M1 polarization and increases the M2 polarization of macrophages (Figure 7D,E). These all data confirm that MKK7 and JNK1/2 have overlapping function in macrophages and inflammation.

## 3. Discussion

The MKK4 and MKK7 pathways are mediators of cytokine signaling, which results in JNK and p38 MAPK activation, leading to the transcriptional activation of cytokine genes and, eventually, cytokine production [3,11]. Biochemical studies using cultured primary cells have provided substantial evidence to support this conclusion [6,11]. However, mice lacking *mkk4* or *mkk7* die [6,8,11,22] before birth, making it difficult to determine the role of MKK4 and MKK7 in vivo. Therefore, there has been limited research investigating the roles of MKK4 and MKK7 in vivo. To overcome this problem, in this study, we generated conditional MKK4 and MKK7 knockout mice to examine the roles and contributions of the MKK4 and MKK7 pathways in macrophage activation and cytokine production in vitro and in vivo.

Dissecting the cell type-specific mechanisms that dominate the innate immune response is critical for the successful management strategies of the uncontrolled inflammatory response [14,23]. Notably, our in vitro and in vivo data showed that, during LPS-induced cytokine production, MKK7 plays a major role in comparison to MKK4. Our data demonstrated that MKK7 regulates cell polarization and migration, an essential feature of efficient macrophage functionality. The ability of MKK7 to control polarization and migration of macrophages suggests that MKK7 can regulate inflammation. We also observed that JNK knockout mimics the LPS-induced cytokine phenotype caused by MKK7 deficiency. Although unlike cytokine production data our polarization data showed that MKK7 knockout varies a bit in the phenotype with JNK1/2 macrophage (Figure 6A,B and Figure 7D,E). It can be explained either by some role of MKK7 in p38 activation or JNK family member JNK3 may present in macrophage and play some role.

MAPK and Nuclear factor-κB (NFκB) pathways both play an essential role in cytokine production [24]. Specifically, both innate immune and adaptive cells, such as macrophages and T-cells produce various pro-inflammatory cytokines in a MAPK-dependent manner during infection, causing an inflammatory response.

JNK requires phosphorylation on both Thr180 and Tyr182 for its complete activation [6]. JNK phosphorylation occurs preferentially on Tyr182 by MKK4 and on Thr180 by MKK7 [11,25]. Additionally, it is known that both MKK4 and MKK7 are required for optimal activation of JNK in fibroblast [6,11]. MKK4 and MKK7 appear to contribute equally to JNK activation in response to UV radiation [6]. However, MKK7 is essential for JNK activation by TNFα, and MKK4 is only required for maximum JNK activation in MEFs [11]. The cooperation between the two kinases, MKK7 and MKK4, may be true in macrophages as well, where MKK7 plays a major role in JNK activation, and MKK4 is only required to maximize the JNK activation. Although the role of MKK7 in the activation of p38 MAPK has not been established, our data showed that MKK7 might play a minor role in p38 MAPK activation as well. We can speculate that similar to JNK activation, p38 MAPK may require MKK7 with its main activator, MKK3/6 [10], for optimal activation in macrophages.

JNK and p38 are long-studied stress signaling cascades essential for a broad range of vital cellular and organismal activities. Therefore, the use of small molecular inhibitors against JNK or p38 to prevent cytokine production and inflammation would likely be accompanied by severe side effects, making them poor drug targets. Several members of the MAP2K and MAP3K tier of kinases were shown to activate JNK and p38 pathways [4,16,26]. Such diversity in the repertoire of upstream activators might, in turn, define the specificity of the stimuli-related pathway activation. Therefore, the data presented in this manuscript suggest that MKK7 plays a vital role in the activation of JNK and the production of TNFα, making it a potential drug target in preventing cytokine production and inflammation.

## 4. Materials and Methods

### 4.1. Animals

Inducible Rosa26 Cre strain mice were obtained from The Jackson Laboratory (004847). Mice with floxed *Mkk4* gene [19], floxed *Mkk7* (exons 3–6 flanked by LoxP site) [12] and floxed *Jnk1 Jnk2−/−* [27] were bred with the inducible Rosa26-Cre^ERT^ mouse line on the C57BL/6J background (Appendix A). Both mice were injected with a single dose of tamoxifen (100 mg/kg body weight in corn oil) for respective gene deletion.

All mouse experiments were performed according to the relevant ethical regulations. Mice were housed in a facility accredited by the American Association for Laboratory Animal Care. All animal studies were approved by the Institutional Animal Care and Use Committee of the University of Massachusetts Medical School and the Brigham Women’s Hospital, Boston, MA, USA.

### 4.2. Cell Culture

Bone marrow-derived primary macrophages (BMDMs) were prepared by flushing the bone marrow cells from the femur and tibia of mice using a 23 G needle. The cells were cultured in a medium consisting of Dulbecco’s modified Eagle’s medium (DMEM) supplemented with 20% heat-inactivated fetal bovine serum (FBS), 2 mM L-glutamine, 1 mM sodium pyruvate, 0.1 mM nonessential amino acids, 100 U/mL penicillin, and 100 ug/mL streptomycin (Life Technologies, Grand Island, NY, USA) in the presence of 30% supernatant from L929 cells as a source of macrophage colony-stimulating factor (M-CSF1; 10–50 ng/mL). After 3–4 days of culture, non-adherent cells were removed, and adherent monolayers were washed in PBS and incubated in fresh media. Experimental cells were resuspended by gently scraping into ice-cold PBS, then cultured in DMEM with 1% FBS, 100 U/mL penicillin, and 0.1 mg/mL streptomycin. Polarization studies were performed using macrophages (7–10 days in culture) incubated with 100 ng/mL IFNγ or 10 ng/mL LPS (M1) for 8 h or 10 ng/mL IL4 (M2) for 24 h.

Primary mouse embryonic fibroblast cells (MEFs) were isolated by dissecting 13.5 days of embryos. After the dissection of embryos, the brain and dark red organs were removed. Using razor blades and a small amount of PBS, embryos were minced. After mincing 5 mL of 0.05% trypsin-EDTA (Gibco) was added to the suspended cells/tissue. The cells/tissue were then incubated at 37 °C, with gentle shaking with 5 mm glass beads, for 15 min. Isolated primary MEFs were cultured in Dulbecco’s modified Eagle’s medium supplemented with 10% fetal bovine serum, 100 U/mL penicillin, 100 ug/mL streptomycin, and 2 mM L-glutamine (Life Technology). 

Both BMDMs and MEFs isolated from different genotyped mice were treated with 4-hydroxytamoxifen (2 uM) for 24 h for respective gene deletion. Cells were used for experiments after 5 days of 4-hydroxytamoxifen treatment.

### 4.3. Cell Migration and Invasion Assays

BMDMs with serum-free cell culture media were plated (100 μL) on top of the filter membrane in a transwell insert and incubated for 10 min at 37 °C and 5% CO_2_ to allow the cells to settle down (Corning, Corning, NY, USA). After 10 min, 600 μL of the DMEM media with monocyte chemoattractant protein 1 (MCP1) chemoattractant (10 uM) were added into the bottom of the lower chamber in a 24-well plate. Transwell inserts were removed from the plate and cleaned using a cotton-tipped applicator. Afterward, the transwell inserts were placed into 70% ethanol for 10 min to allow cell fixation. Cells were incubated in 0.2% crystal violet at room temperature for 5–10 min. After incubation, transwell inserts were gently washed in PBS. An inverted microscope was used to count the migrating cells. For the invasion assay, Matrigel was thawed until liquified on ice, and then 50 µL of Matrigel was added to a 24-well transwell insert and solidified in a 37 °C incubator for 15–30 min to form a thin gel layer before cells were plated [28]. 

### 4.4. RNA Preparation and Quantitative Polymerase Chain Reaction

Quantitative RT–qPCR assays (TaqMan© assays) of mRNA expression were performed using a 7500 Fast Real-Time PCR machine (Applied Biosystems, Waltham, MA, USA) with total RNA prepared from cells with an RNeasy mini kit (Qiagen, Hilden, Germany). TaqMan© assays were used to quantitate *Il6* (Mm00446190_m1), *Tnf*α (Mm00443258_m1), and *GAPDH* (4352339E-0904021) mRNA (Applied Biosystems, Waltham, MA, USA). The 2^−ΔΔ*CT*^ method is used for the relative quantification of genes [29,30]. Reference gene of *GAPDH* was used to normalize the PCRs in each sample.

### 4.5. Immunoblot Analysis

Cell extracts were prepared using triton lysis buffer (TLB buffer) [20 mM Tris (pH 7.4), 1% Triton X-100, 10% Glycerol, 137 mM NaCl, 2 mM EDTA, 25 mM β-Glycerophosphate] with proteinase inhibitors (Sigma #11873580001, St. Louis, MO, USA) and phosphatase inhibitors (Sigma #4906837001, St. Louis, MO, USA). Protein extracts (50 µg of protein) in β-mercaptoethanol containing SDS sample buffer were separated in 4% to 12% gradient SDS-polyacrylamide gels (Bio-Rad #456-8094) and transferred to nitrocellulose membranes (Bio-Rad #170-4271, Hercules, CA, USA) and incubated with a primary antibody with 1:1000 dilution overnight. Membranes were washed in Tris-buffered saline (TBS) with 0.1% Tween-20 for three times and incubated in HRP conjugated secondary antibody for 1 h. Immunocomplexes were visualized with horseradish peroxidase-conjugated secondary antibodies and detected with a Clarity Western ECL substrate (Bio-Rad #170-5061, Hercules, CA, USA). Images were acquired on a chemiluminescent imager (Bio-Rad Chem-Doc Imaging System, Hercules, CA, USA) and quantified using Image J (NIH, Bethesda, MD, USA or Bio-Rad software (Hercules, CA, USA).

### 4.6. Analysis of Blood

Plasma cytokines and cytokines secreted in cell culture media were measured using ELISA (Luminex 200; Millipore, Burlington, MA, USA) by following the manufacturer’s instructions [31].

### 4.7. Antibodies and Reagents

Primary antibodies were obtained from Cell Signaling [MKK4 #9152, MKK7 #4172, phospho- JNK1/2 (Thr183, Tyr185) #4668, p38 #9212, phospho-p38 (Thr180, Tyr182) #9211, and IκBα #4812, Danvers, MA, USA], BD Pharmingen (JNK1/2 #554285, Bedford, MA, USA), and Sigma (α-Tubulin #T5168, St. Louis, MO, USA).

### 4.8. Statistical Analysis

All data are expressed as mean ± SE, and the numbers of independent experiments are indicated. Statistical comparisons were conducted between 2 groups by use of Student t-test or Mann–Whitney U test as appropriate. Multiple groups were compared with either 1-way Kruskal–Wallis or ANOVA with a post hoc Tukey–Kramer multiple comparisons test as indicated in legends. A *p* value < 0.05 was considered significant. All statistics were done using StatView version 5.0 (SAS Institute, Cary, NC, USA) or GraphPad Prism version 5 (GraphPad Software, La Jolla, CA, USA).

## Figures and Tables

**Figure 1 ijms-22-09364-f001:**
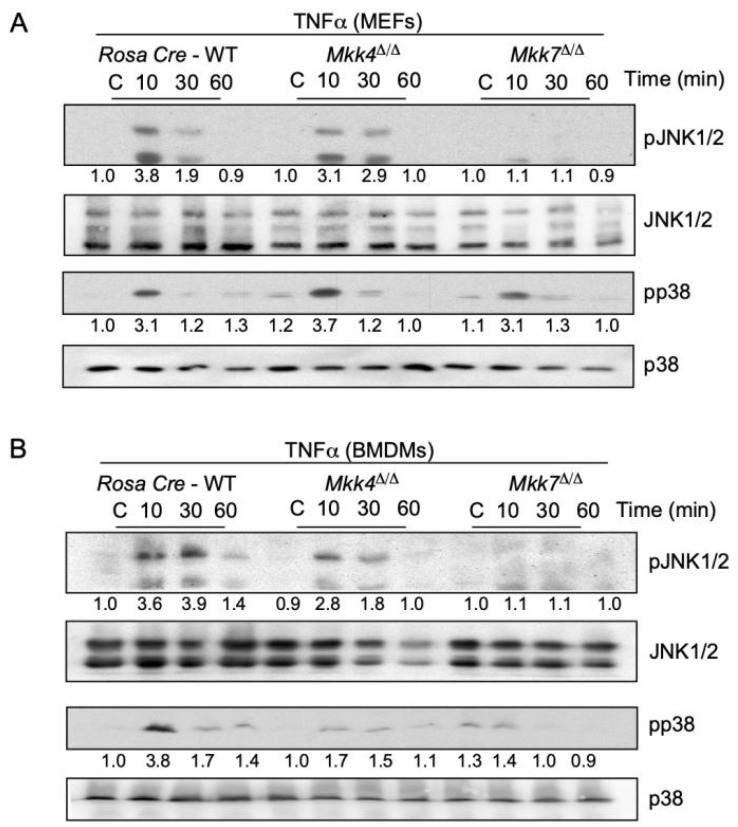
TNFα causes activation of the MKK7 signal transduction pathway. (**A**) Mouse embryonic fibroblast cells (MEFs) isolated from Rosa26-Cre^ERT^, Rosa26-Cre^ERT^-*Mkk4*-floxed, and Rosa26-Cre^ERT^-*Mkk7*-floxed conditional mice were treated with the 4-hydroxytamoxifen for respective gene deletion. 5 days post deletion, MEFs were treated with or without 10 ng/mL TNFα across three-time points (in minutes). MAP kinase JNK and p38 activation were examined by immunoblot analysis. (**B**) Bone marrow drives primary macrophages (BMDMs) from different genotypes were treated with or without 10 ng/mL TNFα across three-time points (in minutes). JNK and p38 MAPK activation were examined by immunoblot analysis. All relative quantification values of western blots were measured in comparison to control. All the experiments were repeated 3–4 times.

**Figure 2 ijms-22-09364-f002:**
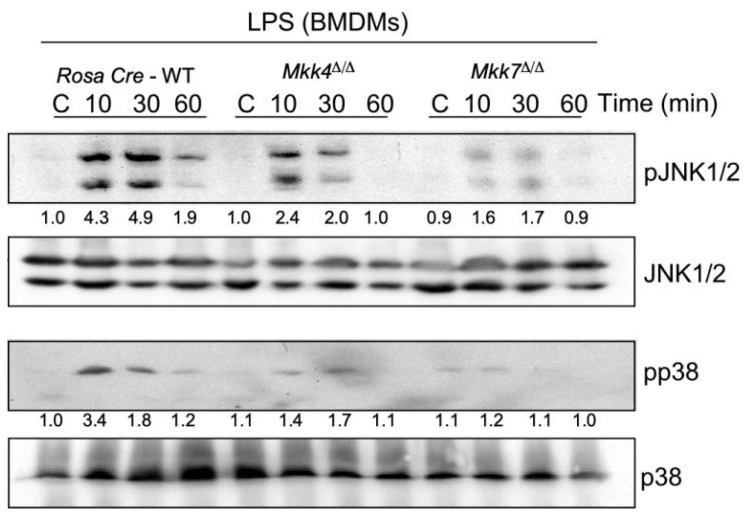
Effect of MKK4 or MKK7-deficiency on the response to LPS. BMDMs isolated from Rosa26-Cre^ERT^, Rosa26-Cre^ERT^-*Mkk4*-floxed, and Rosa26-Cre^ERT^-*Mkk7*-floxed conditional mice were treated with 4-hydroxytamoxifen for respective gene deletion. 5 days post deletion, BMDMs were treated with or without 100 ng/mL LPS for 10, 30, and 60 min. Protein extracts were examined by immunoblot analysis by probing with antibodies to MAP kinases and phospho-MAP kinases. All relative quantification values of western blots were measured in comparison to control. All the experiments were repeated 3–4 times.

**Figure 3 ijms-22-09364-f003:**
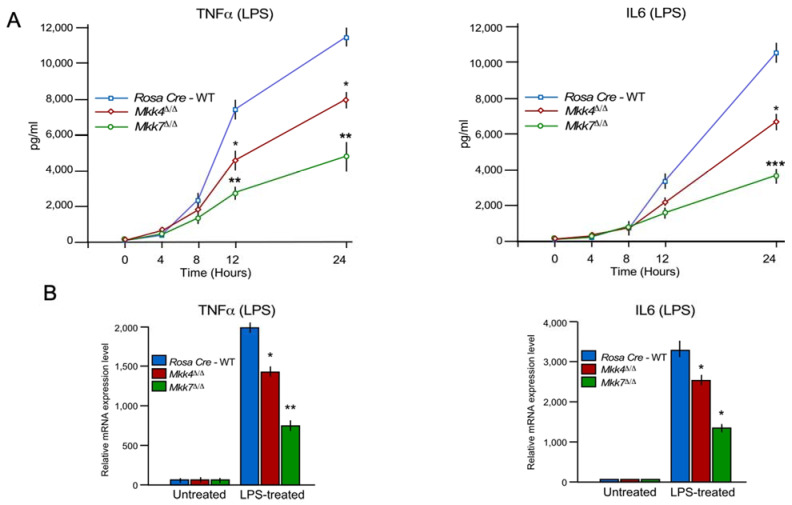
The MKK4 and MKK7 pathway contribute to LPS-stimulated and cytokine production in macrophages. (**A**,**B**) Rosa26-Cre^ERT^, *Mkk4*^∆/∆^ and *Mkk7*^∆/∆^ BMDMs were treated with 100 ng/mL LPS. The amount of TNFα and IL6 in the culture medium was measured by ELISA (**A**), and RT-qPCR was performed to measure mRNA production of TNFα and IL6 (**B**) (mean ± S.E.M.; *n* = 6–8). Statistically significant differences between groups are indicated. (*) *p* < 0.05; (**) *p* < 0.01; (***) *p* < 0.001 vs. WT. All the experiments were repeated 3–4 times.

**Figure 4 ijms-22-09364-f004:**
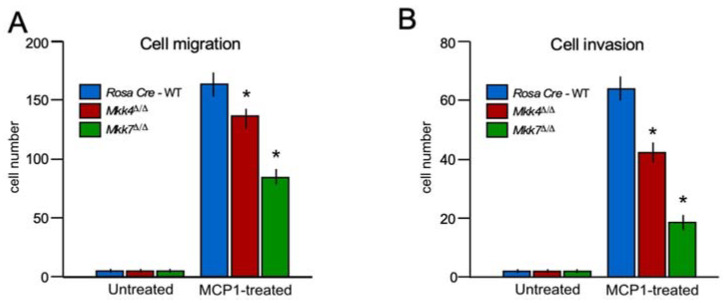
Role of MKK4 and MKK7 in cell migration and invasion. (**A**,**B**) A Transwell system was used to measure the migration (**A**) and Matrigel invasion assay (**B**) of Rosa26-Cre^ERT^, *Mkk4*^∆/∆^ and *Mkk7*^∆/∆^ BMDMs. (mean ± S.E.M.; *n* = 8); (*) *p* < 0.05 vs. WT. All the experiments were repeated 3–4 times.

**Figure 5 ijms-22-09364-f005:**
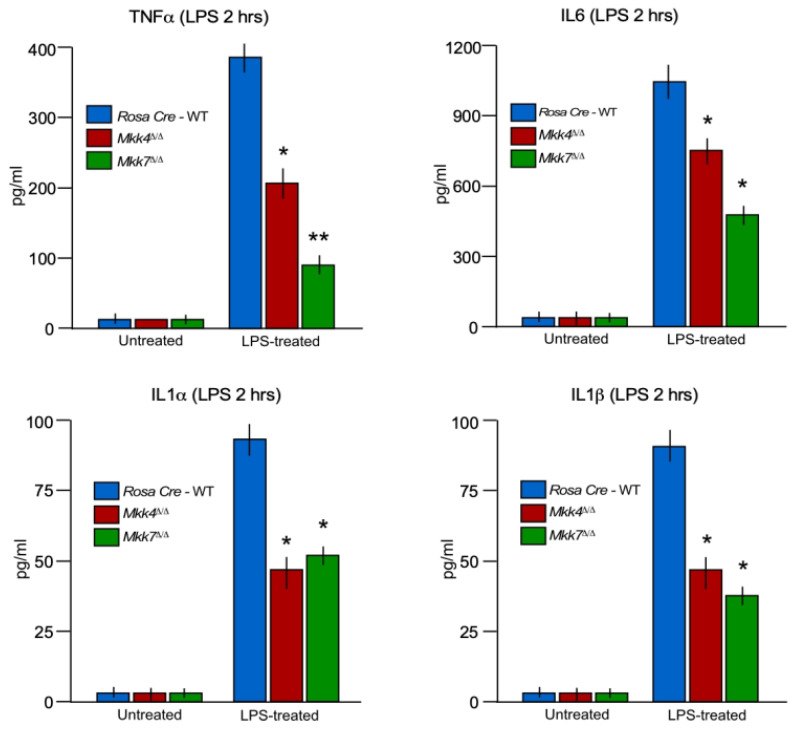
The MKK4 and MKK7 pathway contribute to LPS-mediated inflammation. Rosa26-Cre^ERT^, *Mkk4*^∆/∆^, and *Mkk7*^∆/∆^ mice were treated with or without 20 mg/kg LPS by intraperitoneal injection (2 h). The amount of TNFα, IL6, IL1α, and IL1β in the serum was measured by ELISA (SE; *n* = 8). Statistically significant differences between wild-type and MLK-deficient mice are indicated. (mean ± S.E.M.; *n* = 8); (*) *p* < 0.05; (**) *p* < 0.01 vs. WT. All the experiments were repeated 3–4 times.

**Figure 6 ijms-22-09364-f006:**
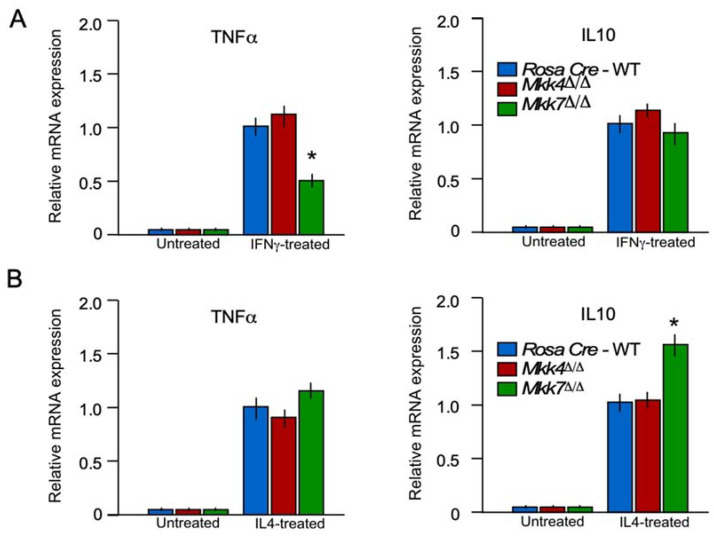
MKK7 is required for macrophages polarization. (**A**) Total RNA was isolated from BMDM incubated (8 h) without or with 100 ng/mL IFNγ. The relative mRNA expression of M1 (TNFα) and M2 (IL10) markers were measured by RT-qPCR assays. (**B**) Total RNA was isolated from BMDM incubated with, or without, 10 ng/mL IL4 (24 h). The relative mRNA expression of the M1 (TNFα) and M2 (IL10) markers were measured by RT-qPCR assays. Statistically significant differences between groups are indicated (mean ± S.E.M.; *n* = 4–6; * *p* < 0.05). All the experiments were repeated 3–4 times.

**Figure 7 ijms-22-09364-f007:**
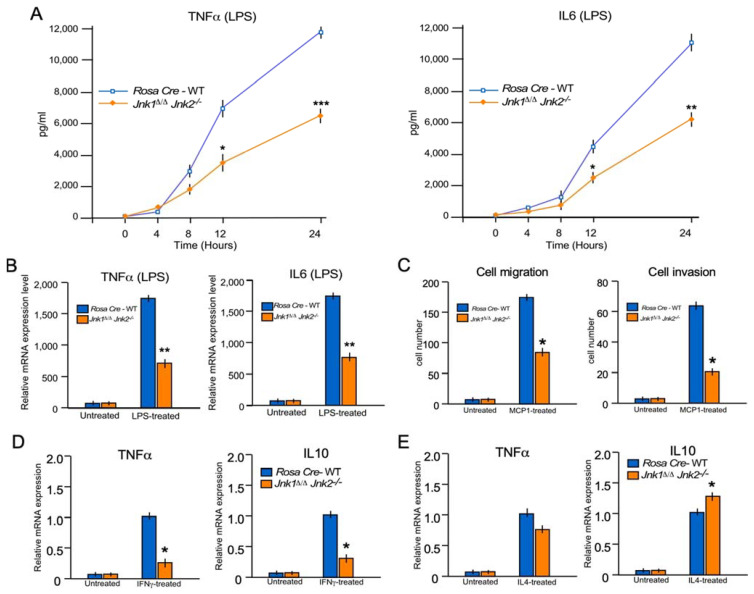
The JNK1/2 mimics the MKK7 phenotype and contributes to LPS-stimulated and cytokine production in macrophages. (**A**,**B**) Rosa26-Cre^ERT^ and *Jnk1*^∆/∆^
*Jnk2^−/−^* BMDMs were treated with 100 ng/mL LPS. The amount of TNFα and IL6 in the culture medium was measured by ELISA (**A**), and RT-qPCR was performed to measure mRNA production of TNFα and IL6 (**B**). (**C**) Rosa26-Cre^ERT^, *Jnk1*^∆/∆^
*Jnk2^−/−^* BMDMs were used to measure the migration and Matrigel invasion assay by the Transwell system. (**D**) Total RNA was isolated from BMDM incubated (8 h) without or with 100 ng/mL IFNγ. The relative mRNA expression of M1 (TNFα) and M2 (IL10) markers were measured by RT-qPCR assays. (**E**) Total RNA was isolated from BMDM incubated without or with 10 ng/mL IL4 (24 h). The relative mRNA expression of the M1 (TNFα) and M2 (IL10) markers were measured by RT-qPCR assays. (mean ± S.E.M.; *n* = 6). Statistically significant differences between groups are indicated. (*) *p* < 0.05; (**) *p* < 0.01; (***) *p* < 0.001 vs. WT. All the experiments were repeated 3–4 times.

## Data Availability

The data sets generated and analyzed as part of this study are available upon request from the corresponding author.

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
