# Peer review of "Mitogen Kinase Kinase (MKK7) Controls Cytokine Production In Vitro and In Vivo in Mice"

_ijms, 2021, doi:10.3390/ijms22179364_

Round 1

Reviewer 1 Report

This is an article describing the role of MKK4 and 7 in the production of pro-inflammatory cytokines. The authors used GEMM to isolate cells and study the impact of these kinases on the production of a number of significant cytokines involved in inflammation. The experimental design is good, the manuscript is well written and the data support clearly the conclusions.

I only have some minor comments that need to be addressed by the authors

-The authors need to define how many mice/independent experiments were performed and what each bar and error bars show in the graphs?

-They must give details on the migration and invasion assays ad on how the BMDM and MEFs were isolated

-Also, more details must be given regarding the Abs: lots and/or clones, incubation times and buffers, blicking conditions etc.

-What are the phospho residues detected in jnk and p38?

Author Response

Dear Editor,

We thank the reviewer for finding our work significant and interesting. The reviewers’ enthusiasm for our work showing MKK7 pathway an important player in to regulate the cytokine and inflammation was much appreciated. The insightful reviewer comments and suggestions resulted in a significantly strengthened manuscript. The changes to each section have been highlighted in the text in yellow.

Reviewer 1:

This is an article describing the role of MKK4 and 7 in the production of pro-inflammatory cytokines. The authors used GEMM to isolate cells and study the impact of these kinases on the production of a number of significant cytokines involved in inflammation. The experimental design is good, the manuscript is well written and the data support clearly the conclusions.

I only have some minor comments that need to be addressed by the authors

-The authors need to define how many mice/independent experiments were performed and what each bar and error bars show in the graphs?

Response: We really thank the reviewer for finding this oversight. Now we have added the information in figure legends including the number of sample size.

-They must give details on the migration and invasion assays ad on how the BMDM and MEFs were isolated

Response: We have extended the method and materials sections to include all the information in the manuscript suggested by reviewer. 

-Also, more details must be given regarding the Abs: lots and/or clones, incubation times and buffers, blicking conditions etc.

Response: We have extended the method and materials section to include this information.

-What are the phospho residues detected in jnk and p38?

Response: We have now provided the information about the phospho residues in the method and materials section

Reviewer 2 Report

In this manuscript, Caliz et al. aim to define the precise pro-inflammatory signaling mechanisms in macrophages that promote stress pathway activation and cytokine production. Through this lens, the authors seek to address the larger question of how different cell types coordinate a response to specific stimuli to produce the appropriate biological ooutcome, such as the innate immune response. Using a mouse model of inducible knockout of MAP2K kinases, Mkk7 or Mkk4, the authors identify Mkk7 as the primary activator of JNK, and to a lesser degree p38, downstream of LPS stimulation with minor contributions from Mkk4. They demonstrate that Mkk7 is necessary for pro-inflammatory cytokine production both in vitro and in vivo. Finally, the demonstrate that Mkk7, and to a lesser extent Mkk4, loss result in diminished capacity for macrophage migration and invasion.

Overall Comments

The authors focus on an important question of how cells appropriately respond to stimuli to produce the necessary biology output. Although the authors provide new mouse models of inducible loss of Mkk7 or Mkk4, the data generated with these models are a minor extension of work in Mkk7 or Mkk4 knockout MEFs from Tournier et al. (2001, G&D). The in vitro experiments are intriguing but limited in impact. To provide a more robust extension of the current knowledge, the authors should take advantage of the in vivo model beyond a single LPS injection experiment. Additional experiments would improve the significance and overall impact of the work.

The Jnk1/2 loss experiments (Fig. 4 and 6) do not provide any additional information about how the Mekk7/4 phenotypes connect to Jnk1/2. Ideally, the authors should conduct genetic epistasis experiments to demonstrate that the Mekk7 and Mekk4 cytokine production and migration/invasion phenotypes are acting through Jnk1/2. That is to say, do Mkk7 or Mkk4 loss plus Jnk1/2 knockout alter the strength of the phenotypes of Mkk7 or Mkk4 loss alone? In the current state, the Jnk1/2 knockout experiments are nice positive controls, but do not add any new information. The rationale provided for the Jnk1/2 knockdout experiments should be made clearer or ideally improved through epistasis studies.

The migration and invasion experiments provide compelling data that Mekk7 and Mekk4 are important for the cellular phenotypic response to immune stimuli. However, there is no description of how the experiments were conducted in either the main text or the methods. Were the cells migrating/invading towards a specific pro-inflammatory stimulus? This must be stated such that the reader can understand how or if this cellular phenotype corresponds to the changes in mRNA/protein production after LPS stimulation.

Specific Comments

  1. Fig S1E- Are the cells used for this experiment BMDMs as stated in the main manuscript text or MEFs as stated in the figure legend?
  2. 1 & 2- The pp38 blots are of quite poor quality thus making data interpretation difficult.
  3. 3 & 4- The x-axis groups on the bar graphs should be labeled, e.g. untreated or treated. Particularly, in Fig. 4 it is unclear what is the control group condition.
  4. The title to section 2.6 should not state anything about “in vivo” results as all JNK1/2 loss experiments are done in vitro.

Author Response

Dear Editor,

We thank the reviewer for finding our work significant and interesting. The reviewers’ enthusiasm for our work showing MKK7 pathway an important player in to regulate the cytokine and inflammation was much appreciated. The insightful reviewer comments and suggestions resulted in a significantly strengthened manuscript. The changes to each section have been highlighted in the text in yellow.

Reviewer 2:

In this manuscript, Caliz et al. aim to define the precise pro-inflammatory signaling mechanisms in macrophages that promote stress pathway activation and cytokine production. Through this lens, the authors seek to address the larger question of how different cell types coordinate a response to specific stimuli to produce the appropriate biological ooutcome, such as the innate immune response. Using a mouse model of inducible knockout of MAP2K kinases, Mkk7 or Mkk4, the authors identify Mkk7 as the primary activator of JNK, and to a lesser degree p38, downstream of LPS stimulation with minor contributions from Mkk4. They demonstrate that Mkk7 is necessary for pro-inflammatory cytokine production both in vitro and in vivo. Finally, the demonstrate that Mkk7, and to a lesser extent Mkk4, loss result in diminished capacity for macrophage migration and invasion.

Overall Comments

The authors focus on an important question of how cells appropriately respond to stimuli to produce the necessary biology output. Although the authors provide new mouse models of inducible loss of Mkk7 or Mkk4, the data generated with these models are a minor extension of work in Mkk7 or Mkk4 knockout MEFs from Tournier et al. (2001, G&D). The in vitro experiments are intriguing but limited in impact. To provide a more robust extension of the current knowledge, the authors should take advantage of the in vivo model beyond a single LPS injection experiment. Additional experiments would improve the significance and overall impact of the work.

 Response: We totally agree with the reviewer referencing this manuscript to follow up Tournier et al. (2001, G&D). But as whole body MKK7 and MKK4 knockout mice were lethal, we have developed inducible conditional knockout mice. We have provided the new data about the role of MKK4 and MKK7 in innate immunity. Additionally, we have now added new experimental results (Figure 6) which show that MKK7 is required for macrophage polarization. This may also explain why we see the difference in cytokine production in vitro and in vivo.

The Jnk1/2 loss experiments (Fig. 4 and 6) do not provide any additional information about how the Mekk7/4 phenotypes connect to Jnk1/2. Ideally, the authors should conduct genetic epistasis experiments to demonstrate that the Mekk7 and Mekk4 cytokine production and migration/invasion phenotypes are acting through Jnk1/2. That is to say, do Mkk7 or Mkk4 loss plus Jnk1/2 knockout alter the strength of the phenotypes of Mkk7 or Mkk4 loss alone? In the current state, the Jnk1/2 knockout experiments are nice positive controls, but do not add any new information. The rationale provided for the Jnk1/2 knockdout experiments should be made clearer or ideally improved through epistasis studies.

 Response: Reviewer is right in  suggesting these loss of function experiments. Although making triple knockout mice is a time-consuming process, we will proceed with these nice suggestions for our future work.

The migration and invasion experiments provide compelling data that Mekk7 and Mekk4 are important for the cellular phenotypic response to immune stimuli. However, there is no description of how the experiments were conducted in either the main text or the methods. Were the cells migrating/invading towards a specific pro-inflammatory stimulus? This must be stated such that the reader can understand how or if this cellular phenotype corresponds to the changes in mRNA/protein production after LPS stimulation.

 Response: We really thank the reviewer for finding this oversight. We have added the information into the manuscript as suggested by the reviewer. 

Specific Comments

  1. Fig S1E- Are the cells used for this experiment BMDMs as stated in the main manuscript text or MEFs as stated in the figure legend?

Response: We really thank the reviewer for finding this oversight. We have now corrected this mistake.

  1. 1 & 2- The pp38 blots are of quite poor quality thus making data interpretation difficult.

Response: Reviewer is right to  cite the quality as the pp38 antibody is difficult to work with in comparison to pJNK and pERK antibodies. Nonetheless to clarify the results, we have used densitometry analysis to measure the difference in band intensity.

  1. 3 & 4- The x-axis groups on the bar graphs should be labeled, e.g. untreated or treated. Particularly, in Fig. 4 it is unclear what is the control group condition.

Response: We really thank the reviewer for this suggestion. We have now added the group names as suggested. We also have provided information in the method and materials section referencing  treatment conditions.

  1. The title to section 2.6 should not state anything about “in vivo” results as all JNK1/2 loss experiments are done in vitro

Response: We have changed the subheading as suggested by the reviewer.

Round 2

Reviewer 2 Report

Overall Comments

The authors have sufficiently modified the manuscript in response to the original comments. The new macrophage polarization studies provide significant new data that expand the impact of the study and provide new insights into the role of MAP2K/JNK signaling during the inflammatory response.

A few minor points still need to be addressed:

  1. The quantification of the western blots in Figs. 1 & 2 is appreciated. However, it should be made clear what the values are being compared to (e.g. to WT-control?).

  1. The flow and clarity of the paper would be improved by either putting all the Jnk1/2 knockout experiments with their matched Mkk 4 or 7 knockout experiments. That is to say, section 2.7 should be combined into section 2.3 (add Fig. 7 to Fig. 3). Otherwise, all the Jnk1/2 knockout data, including the migration data from Fig. 4, should move to section 2.7. As it is currently written it doesn’t make sense why Jnk1/2 is being revisited at the end of the manuscript.

  1. While the new macrophage polarization data provides originality to the study, the authors should include comments regarding the stark contrast in phenotypes between Mkk4 and Mkk7 loss that are much more striking than the results of all other assays. Also, it is curious that Jnk1/2 loss doesn’t fully mimic Mkk7 loss in this assay.

Author Response

Dear Editor,

We would like to thank again to the reviewer for his/her helpful comments. We found reviewer comments insightful which resulted in a significantly strengthened manuscript. The new changes to each section have been highlighted in the text in turquoise.

Overall Comments

The authors have sufficiently modified the manuscript in response to the original comments. The new macrophage polarization studies provide significant new data that expand the impact of the study and provide new insights into the role of MAP2K/JNK signaling during the inflammatory response.

A few minor points still need to be addressed:

  1. The quantification of the western blots in Figs. 1 & 2 is appreciated. However, it should be made clear what the values are being compared to (e.g. to WT-control?).

Response: We really thank the reviewer for finding this oversight. Now we have added the information in figure legends.

  1. The flow and clarity of the paper would be improved by either putting all the Jnk1/2 knockout experiments with their matched Mkk 4 or 7 knockout experiments. That is to say, section 2.7 should be combined into section 2.3 (add Fig. 7 to Fig. 3). Otherwise, all the Jnk1/2 knockout data, including the migration data from Fig. 4, should move to section 2.7. As it is currently written it doesn’t make sense why Jnk1/2 is being revisited at the end of the manuscript.

Response: We totally agree with the reviewer and have moved all data JNK1/2 knockout data section 2.7.

  1. While the new macrophage polarization data provides originality to the study, the authors should include comments regarding the stark contrast in phenotypes between Mkk4 and Mkk7 loss that are much more striking than the results of all other assays. Also, it is curious that Jnk1/2 loss doesn’t fully mimic Mkk7 loss in this assay.

Response: Reviewer is right in suggesting that even though MKK7 and and JNK1/2 may have overlapped function there are some degree of variation we saw in polarization experiment. We have now included it into discussion.
